# Cytochrome P450 Expression and Chemical Metabolic Activity before Full Liver Development in Zebrafish

**DOI:** 10.3390/ph13120456

**Published:** 2020-12-11

**Authors:** Tasuku Nawaji, Natsumi Yamashita, Haruka Umeda, Shuangyi Zhang, Naohiro Mizoguchi, Masanori Seki, Takio Kitazawa, Hiroki Teraoka

**Affiliations:** 1School of Veterinary Medicine, Rakuno Gakuen University, 582, Bunkyodai-Midorimachi, Ebetsu, Hokkaido 069-8501, Japan; gabugabuwolf@gmail.com (N.Y.); s21661138@stu.rakuno.ac.jp (H.U.); s21741002@stu.rakuno.ac.jp (S.Z.); tko-kita@rakuno.ac.jp (T.K.); 2Chemicals Evaluation and Research Institute, Japan (CERI), 3-2-7, Miyanojin, Kurume, Fukuoka 839-0801, Japan; mizoguchi-naohiro@ceri.jp (N.M.); seki-masanori@ceri.jp (M.S.)

**Keywords:** cytochrome P450, drug metabolism, developing zebrafish

## Abstract

Zebrafish are used widely in biomedical, toxicological, and developmental research, but information on their xenobiotic metabolism is limited. Here, we characterized the expression of 14 xenobiotic cytochrome P450 (CYP) subtypes in whole embryos and larvae of zebrafish (4 to 144 h post-fertilization (hpf)) and the metabolic activities of several representative human CYP substrates. The 14 CYPs showed various changes in expression patterns during development. Many CYP transcripts abruptly increased at about 96 hpf, when the hepatic outgrowth progresses; however, the expression of some *cyp1*s (*1b1*, *1c1*, *1c2*, *1d1*) and *cyp2r1* peaked at 48 or 72 hpf, before full liver development. Whole-mount in situ hybridization revealed *cyp2y3*, *2r1*, and *3a65* transcripts in larvae at 55 hpf after exposure to rifampicin, phenobarbital, or 2,3,7,8-tetrachlorodibenzo-*p*-dioxin from 30 hpf onward. Marked conversions of diclofenac to 4′-hydroxydiclofenac and 5-hydroxydiclofenac, and of caffeine to 1,7-dimethylxanthine, were detected as early as 24 or 50 hpf. The rate of metabolism to 4’-hydroxydiclofenac was more marked at 48 and 72 hpf than at 120 hpf, after the liver had become almost fully developed. These findings reveal the expression of various CYPs involved in chemical metabolism in developing zebrafish, even before full liver development.

## 1. Introduction

In recent years, zebrafish (*Danio rerio*) embryos and larvae have been used as alternative in vivo toxicity screening models early in the drug discovery process because of their low cost, the need for only small amounts of test drugs, and their high throughputs [1,2,3,4]. Moreover, because in Europe zebrafish embryos are considered non-protected animals until the stage of independent feeding at 120 h post-fertilization (hpf) based on directive on the protection of animals used for scientific purposes, their use is in line with the 3Rs (reduce, refine, and replace) approach to animal use for scientific purposes [5]. Several phenotype-based alternative methods for examining developmental toxicity by using zebrafish embryos have been reported and have shown high concordance (81% to 90%) with the findings of in vivo mammalian studies [6,7,8,9].

Despite these advances, a problem remains in achieving more accurate predictions of the toxicity of various drugs to humans and other mammals: We still have insufficient information about the uptake and metabolism of drugs in zebrafish embryos and larvae. With regard to drug uptake, although most developmental toxicity evaluations by using the above mentioned phenotype-based methods have been based on the concentration of each compound in water (C_w_), the C_w_ does not reflect the amount of toxicant that is directly associated with developmental toxicity [10]. Only a few papers have reported the contents of compounds in whole zebrafish embryos or larvae (C_e_) [10,11,12,13]; further data on the C_e_ for various compounds are therefore needed to determine the uptake properties of these drugs in zebrafish embryos or larvae in detail. With respect to metabolism, zebrafish embryos and larvae depend on their own drug-metabolizing capacity for detoxification of xenobiotics because of the lack of a maternal barrier, whereas mammalian embryos and fetuses are exposed to the parent compound and its metabolites via the maternal metabolism. In particular, the presence or absence of direct drug-metabolizing capacity is important in studies of proteratogens, which require bioactivation to exert their teratogenic potential [14]. A lack of intrinsic capacity in zebrafish to metabolize proteratogens may lead to false negative results in teratogenicity assays of these compounds. Despite these potential issues, there have been few comparisons of drug metabolism between zebrafish embryos or larvae and humans or other mammals [12].

Drug metabolism occurs mainly through cytochrome P450 enzymes (CYPs). CYPs are a superfamily of hemoproteins, among which the CYP1, CYP2, and CYP3 families participate largely in the oxidative metabolism of xenobiotics [14,15,16]. CYPs account for approximately 75% of the enzymes involved in the metabolism of marketed pharmaceuticals [17]. Of the 57 CYPs encoded in the human genome, a mere five CYP family members (CYP1A2, CYP2C9, CYP2C19, CYP2D6, and CYP3A4/5) account for approximately 95% of the metabolism by CYPs [17]. In the human liver, which is crucial to homeostasis and the protection of individuals against xenobiotics, the three most commonly expressed CYPs are CYP1A2, CYP2C9 and CYP3A4/5, the respective contents of which as percentages of total CYP are 12% to 13%, 12% and 29% to 30% [18,19,20,21].

In zebrafish, our knowledge of CYP-mediated drug metabolism is limited and fragmented [2,22]. Previous reports have shown that formation of the hepatic primordium in zebrafish begins at 28 hpf; hepatic outgrowth begins between 60 and 72 hpf, and liver function, including CYP metabolism, is almost complete by 120 hpf [23,24,25]. Transcriptional profiling of a full complement of CYP genes in the zebrafish embryo during the first 48 h of development and in the adult zebrafish liver has been reported [26,27]. However, data on the expression of CYP genes in zebrafish from 48 hpf onwards have not yet been reported. Additionally, data on the activity and organ distribution of CYP transcripts in zebrafish remain scarce. Because some drugs activate CYP-mediated metabolism by inducing the biosynthesis of CYP isozymes through particular signaling pathways, metabolic activity in zebrafish exposed to substrates is likely to be different from that in unexposed zebrafish [28]. Therefore, to understand completely CYP-mediated drug metabolism in developing zebrafish embryos and larvae we need to examine the profiles of CYP activity and expression.

Our objective here was to research the profiles of mRNA expression, organ distribution, and metabolic activity in developing zebrafish embryos and larvae for various drug-metabolizing CYPs and to obtain additional data on the uptake of drugs. To meet this objective we performed the following experiments: (1) we used quantitative real-time polymerase chain reaction (qPCR) to analyze the transcript levels of drug-metabolizing CYPs in whole embryos or larvae in comparison with those in the adult zebrafish liver; (2) we performed whole-mount in situ hybridization to analyze the distribution of CYP transcripts in larval zebrafish treated with chemical inducers from before the start of hepatic outgrowth; (3) we measured the C_e_ values of some compounds to obtain more data on their uptake in zebrafish embryos and larvae; and (4) we measured human CYP-mediated metabolites in zebrafish embryos and larvae exposed to CYP substrates to investigate similarities in metabolic function between humans and zebrafish embryos or larvae. In experiments (3) and (4), three drugs, namely caffeine, diclofenac sodium salt (diclofenac), and testosterone, were used for exposure of zebrafish embryos and larvae. The log K_ow_ (the logarithm of the *n*-octanol–water partition coefficient) values for caffeine, diclofenac, and testosterone are −0.07, 1.13 (pH 7.4), and 3.32, respectively [29,30,31]. For experiment (4), we used caffeine, diclofenac, and testosterone as substrates for three human CYPs (CYP1A2, CYP2C9, and CYP3A4/5, respectively) that are considered key in human drug metabolism. In humans, the formation of 1,7-dimethylxanthine is known to be mediated by CYP1A2, and that of 4′-hydroxydiclofenac (4′-OHDF) is mediated by CYP2C9 [32,33]. The CYP3A4/5 pathway mediates the formation of 5-hydroxydiclofenac (5-OHDF) and 6β-hydroxytestosterone in humans [34,35].

## 2. Results

### 2.1. Expression of Metabolic CYP Isoforms in Whole Zebrafish Embryos and Larvae from 4 to 144 hpf

Fourteen CYPs that are homologous to human xenobiotic metabolic CYP isoforms showed various changes in their mRNA expression patterns during development (Figure 1, Figure 2 and Figure 3). Expression of many CYP transcripts abruptly increased at about 96 hpf, when the hepatic outgrowth progresses; however, the expression of some *cyp1s* (*1b1*, *1c1*, *1c2*, *1d1*) (Figure 1B–E) and *cyp2r1* (Figure 2D) peaked at 48 or 72 hpf, before full liver development (Figure 1 and Figure 2). The transcript levels of many CYPs, namely *cyp1b1, 1c1*, *2n13*, *3a65*, *3c2/3*, and *3c4* (Figures A,C,D), in whole embryos or larvae at various points in the period up until 144 hpf were significantly (*p* < 0.05) higher than those in male or female adult livers. Expression levels of *cyp1c2* and *1d1* were not always significantly higher in male or female adult livers than in larvae at some stages (Figure 1D,E). Whole-larval expression of *cyp3a65* was much greater than that of *cyp3c1* and other *cyp3c* subtypes from 96 to 144 hpf, whereas *cyp3c1* dominated over the other *cyp3* subtypes in the adult livers (Figure 3).

### 2.2. Spatial mRNA Expression of CYP Isoforms in Larvae at 55 hpf

To address the localization of the CYP isoforms studied in the previous section, we performed whole-mount in situ hybridization in 55 hpf larvae, before the start of hepatic outgrowth. Positive signals were rarely detected in vehicle-treated larvae (Figure 4E,G,J,N), except in the minor case of *cyp2y3* (Figure 4K), supporting the specificity of RNA probes used.

As positive expression of CYP subtypes studied was rarely detected in the vehicle control larvae, embryos were exposed to CYP inducers (rifampicin, phenobarbital, or 2,3,7,8-tetrachlorodibenzo-*p*-dioxin (TCDD)) [28] from 30 hpf onwards and fixed for in situ hybridization at 55 hpf. *cyp2N13* mRNA expression was detected in larvae exposed to rifampicin (Figure 4E,F) or phenobarbital (data not shown). Localization was comparable to that of *elov1a*, an air bladder primordium marker (A, B). Positive signals of *cyp2r1* mRNA from a location similar to that of the liver, as revealed by *foxa3* mRNA expression (C, D), were recognized in larvae exposed to TCDD (H), phenobarbital (I), or rifampicin (data not shown). *cyp2y3* mRNA was detectable in larvae exposed to rifampicin (L) or phenobarbital (M), as well as in a few vehicle-treated control larvae (K). The expression sites of *cyp2y3* mRNA were similar to the locations of the liver and intestine, as revealed by comparison with *foxa3* mRNA expression (C, D). A *cyp3a65* mRNA signal appeared in some TCDD- or rifampicin-exposed larvae (O, P); the expression sites were similar to those of *foxa3* (C, D) and *cyp2y3* (L, M) mRNAs.

In contrast, in 55 hpf larvae, significant inductions of *cyp2n13*, *2r1*, *2y3*, and *3a65* by rifampicin, and β-naphthoflavone (a TCDD-like CYP inducer [36]) were not recognized by qPCR (Appendix A).

### 2.3. Content of Test Compounds in one Whole Zebrafish Embryo or Larva Every 24 hpf

Every 24 hpf we measured the C_w_ values of the test compounds used to quantify the C_e_ values (Table 1). The time-weighted mean measured C_w_ values were, for caffeine, 10.2 mg/L (102% of the nominal concentration); for diclofenac, 3.41 mg/L (107% of the nominal concentration); and for testosterone, 2.80 mg/L (93.3% of the nominal concentration). The measured C_w_ value of each test compound was therefore close to the nominal value throughout the study.

The C_e_ of caffeine increased steadily and was still increasing at the end of sampling at 120 hpf (Figure 5A). The maximum amount of caffeine per one whole embryo or larva during exposure was 4.54 ± 0.10 ng at 120 hpf. The C_e_ of diclofenac per embryo or larva showed a temporal behavior different from that of caffeine. It reached a maximum value (1.64 ± 0.15 ng) at 72 hpf, but by 120 hpf had gradually and significantly (*p* < 0.05) decreased to 0.285 ± 0.065 ng (Figure 5B). The C_e_ of testosterone per embryo or larva had a temporal behavior similar to that of diclofenac. It reached a maximum value (12.8 ± 1.14 ng) at 48 hpf, but by 120 hpf had gradually decreased to 1.43 ± 0.11 ng (Figure 5C).

### 2.4. Contents of Test Compound Metabolites per Whole Zebrafish Embryo or Larvae

During exposure to caffeine or diclofenac, the respective metabolites 1,7-dimethylxanthine or 4′-OHDF and 5-OHDF were detected and quantified in embryos or larvae both before and after the start of liver function (Figure 6). The C_e_ of caffeine per whole embryo or larva was 2.69 ± 0.14 ng at 50 hpf and 5.15 ± 0.31 ng at 120 hpf (Figure 6A). The C_e_ of the caffeine metabolite 1,7-dimethylxanthine per whole embryo or larva was 0.0161 ± 0.0025 ng at 50 hpf and 0.0355 ± 0.0069 ng at 120 hpf (Figure 6B). The C_e_ of 4′-OHDF per whole embryo or larva showed a temporal behavior similar to that of diclofenac. It reached a maximum value (8.90 ± 0.21 ng/larva) at 72 hpf, but by 120 hpf it had gradually and significantly (*p* < 0.05) decreased to 1.34 ± 0.52 ng (Figure 6C). The C_e_ of 5-OHDF per whole embryo or larva reached a maximum value (2.80 ± 0.31 ng) at 96 hpf, but by 120 hpf it had decreased to 0.601 ± 0.178 ng (Figure 6C). 6β-Hydroxytestosterone was not detected in embryos or larvae at 50 or 120 hpf during exposure to testosterone (data not shown).

## 3. Discussion

Our results revealed the transcriptional changes in 14 CYP isoforms, most of which are homologous to human metabolic CYP isoforms [27] in the whole zebrafish body from just after fertilization to 144 hpf, in comparison with the expression of these isoforms in the zebrafish adult liver. This period is the one most commonly used for pharmacological and toxicological experiments in this species. The zebrafish embryo or larva is an ideal model for such studies of whole body metabolism because of the fish’s very small size. Overall, our data are comparable to those of Goldstone et al. [27], who used a non-statistical microarray analysis to show transcript levels in embryos and larvae for up to 48 hpf. However, *cyp2r1* had a clearly sharp peak at 24 hpf in their microarray study, whereas in our qPCR study its expression level was fairly constant between 4 and 48 hpf but had risen very slightly by 72 hpf (Figure 2D). We found that the transcript levels of some CYP subtypes—*cyp2ad2*, *2k18*, *2n13*, *3a65*, *3c/3*, and *3c4* tended to rise from 96 hpf, while xenobiotic metabolism are fully operational by 5 dpf [25], suggesting that these CYP subtypes were localized in the liver. Nevertheless, the expression of *cyp2n13* in the whole embryo or larva was higher than, or comparable to, that in the adult liver (Figure 2C). Additionally, *cyp3a65* expression in the livers of adults of both sexes was significantly lower than in the whole larva at 120 and 144 hpf (Figure 3A). It is well known that CYP expression profiles in the liver differ greatly between the adult and fetal livers of humans and rodents [39]. Whereas CYP1A2, 2C, 2E, and 3A4 predominate in the human adult liver, 3A7 predominates in the human fetal liver and CYP1A2, 2C, 2D6, and 3A5 are most common in the newborn liver [40]. Our study suggests that conversion of major CYP subtypes between the larval and adult stages also occurs in fish species. Notably, *cyp3a65* was a dominant *cyp3* subtype in whole larvae in the later larval stages (96 to 144 hpf) and *cyp3c1* was a dominant *cyp3* subtype in the adult livers of both sexes. We previously showed by using an RNA sequencing technique that *cyp3a65* is more abundant than other *cyp3* subtypes in the adult zebrafish livers of both sexes; we compiled comprehensive expression profiles of various CYP subtypes in the livers of the adult RIKEN WT (wild-type) line [26]. However, if we are to make final conclusions, further experiments will be needed using different zebrafish lines that are fed with the same food.

We also found that the mRNA expression of most *cyp1* subtypes (*cyp1b1*, *1c1*, *1c2*, and *1d1*) and some *cyp2* subtypes (*cyp2r1*, *2y3*) at 48 and 72 hpf was comparable to, or higher than, that between 96 and 144 hpf. At various points in the embryonic and larval stages up to 144 hpf, the mRNA expression of *cyp2n13*, *3a65* and those CYP subtypes mentioned in the previous sentence (except *cyp2r1*) was comparable to, or even higher than, that in the adult liver.

Expression levels of the mRNAs encoding most metabolic CYP isoforms closely reflect the amounts of the proteins in the human liver, except, for example, in the case of CYP2Es [41]. For CYP2Es, there are relatively large amounts of mRNA but minor amounts of protein in mammals [42]. There is no information on developing zebrafish on relative CYP isoform mRNA expression and protein production as far as we know. Comprehensive studies of the protein abundance of metabolic CYP isoforms in zebrafish need to be performed in the future.

By using the whole-mount in situ hybridization technique we were able to recognize positive signals of *cyp2n13*, *2r1*, *2y3*, and *3a65*, although we rarely detected positive expression of these CYP subtypes in the vehicle control larvae. Positive signals of these CYP subtypes were detected in what were possibly the primordia of the air bladder, liver, and intestine in larvae exposed to CYP inducers, although their inductions were not significant in the qPCR study (Appendix A), possibly because of the very restricted nature of the induction sites. The primordial liver bud and the primordial intestine form at about 36 hpf [43]. We previously reported in 50 hpf larvae during TCDD exposure that mRNA expression of *cyp1c1/2* was restricted to branchiogenic primordia and pectoral fin buds, whereas transcripts of *cyp1a* were localized in the skin and vasculature throughout the body [36,44]. *cyp1b1* is constitutively transcribed in the retina, at the midbrain–hindbrain boundary, and in the diencephalon region [45]. Therefore, many CYP subtypes are expressed not only in the possible primordial liver bud but also in the extrahepatic tissues in zebrafish embryos or larvae. The primordial intestine and the primordial bladder, as well as primordial liver before full liver development, emerged as important extrahepatic CYP expression sites in our study.

Further data on C_e_ values are needed to help us understand the uptake patterns of various compounds in zebrafish embryos and larvae. We therefore analyzed time-series of the C_e_ values of caffeine, diclofenac, and testosterone. These values showed different temporal behaviors depending on the drugs’ lipophilicities. Caffeine is highly water soluble, and its C_e_ increased linearly and was still doing so at 120 hpf. Conversely, the C_e_ values of diclofenac and testosterone, which are highly fat soluble, peaked at 72 and 48 hpf, respectively, and then gradually decreased; they were still decreasing at 120 hpf. These temporal behaviors were almost the same as the previously reported C_e_ values of highly water soluble compounds (caffeine and valproate sodium salt), and a highly fat soluble compound (diethylstilbestrol) in aqueous solution concentrations that caused developmental abnormalities [10]. In our previous report, we surmised that the gradual decline in the C_e_ of highly fat-soluble compounds is caused by a decline in the total lipid concentration in whole embryos or larvae with aging because of the energetic costs of growth and development [10]. We consider that the declines observed here had the same causes. Moreover, we observed no peaks other than those of diclofenac and testosterone on the chromatogram. It is therefore unlikely that the declines in the C_e_ values of diclofenac and testosterone were caused by the biotransformation of these compounds.

To confirm that zebrafish embryos and larvae before full liver development have metabolic activities that are similar to those in humans we exposed zebrafish embryos and larvae to the substrates of three human CYP isoforms and then quantified the substrates and their metabolites. We detected 1,7-dimethylxanthine (a metabolite of caffeine via human CYP1A2) and 4′-OHDF and 5-OHDF (metabolites of diclofenac via human CYP2C9 and CYP3A4/5, respectively) in zebrafish embryos or larvae before and after the time of complete formation of the liver. Although there have been no previous data on the detection of 1,7-dimethylxanthine in zebrafish embryos, larvae, or adults, Alderton et al. reported that human CYP1A2-mediated metabolites of tacrine and phenacetin were detected 7 days post-fertilization (dpf) in zebrafish incubated with those drugs [46]. In a previous paper, the two metabolites of diclofenac were detected in adult zebrafish liver microsomes and in the microsomes of two of five batches of whole zebrafish larvae, but only at 96 hpf: no metabolites were detected in the microsomes at 5, 24, 48, 72, or 120 hpf [22]. However, our embryos and larvae were exposed continuously to diclofenac until sampling to measure C_e_, whereas in the previous report the microsomes were exposed to diclofenac for only 1 h [22]. These differences in the conditions of exposure to diclofenac may have resulted in the differences between our results and those of this other study. Therefore, a novel finding of our study is that zebrafish embryos and larvae in early development are likely to have metabolic functions similar to those of human CYP1A2, CYP2C9, and CYP3A4/5 without depending on liver organogenesis.

Quantification of the C_e_ values of 1,7-dimethylxanthine, 4′-OHDF, and 5-OHDF indicated differences in the properties of their metabolic activities in zebrafish embryos and larvae. The C_e_ of 1,7-dimethylxanthine was only about 5% of that of caffeine at both 50 and 120 hpf, suggesting that zebrafish embryos and larvae have metabolic activity similar to that of human CYP1A2 but that the activity is low. This trend of low metabolic activity is in agreement with the findings of an ethoxyresorufin-O-deethylase assay of Cyp1A activity in zebrafish embryos and larvae [47]. Conversely, in zebrafish embryos and larvae, metabolic activity similar to human CYP2C9- or CYP3A4/5-mediated metabolism is likely to be high, because the maximum C_e_ values of 4′-OHDF and 5-OHDF were 543% and 171%, respectively, of that of diclofenac. To our knowledge, this is the first report of the time-series of C_e_ values of 4′-OHDF and 5-OHDF every 24 hpf up until 120 hpf in zebrafish embryos and larvae during exposure to diclofenac. The temporal behaviors of these values were similar in that they peaked at 72 hpf (4′-OHDF) or 96 hpf (5-OHDF) and then declined by 120 hpf. In general, in enzymatic reactions, the higher the concentration of the substrate the greater the production of the metabolite becomes. Moreover, dynamic changes in the mRNA expression levels of CYP isoforms mediating the metabolism of diclofenac may also influence the temporal behavior of the C_e_ values of 4′-OHDF and 5-OHDF. Therefore, we consider that the temporal behavior of the C_e_ values of these two metabolites was the combined result of changes in the diclofenac C_e_ values and changes in the expression levels of the CYP isoforms mediating diclofenac metabolism. The developmental changes in the expression of *cyp1c1*, *1c2* and *2r1* were similar to the time-courses of the metabolic activities of 4′-OHDF and 5-OHDF, although the transcription levels of *cyp2r1* were much lower than those of the others. Although some CYP1 isoforms, CYP3A65 and CYP3C1, have been relatively well studied in yeast (*Saccharomyces cerevisiae*) expression system, only limited information is available on their metabolic activity with substrates other than caffeine and diclofenac [48,49,50]. In the future, additional experiments, such as in vitro assays with systems of heterogeneous expression of zebrafish CYP isoform proteins and diclofenac using *Escherichia coli* or yeast, would reveal which isoforms play key roles in the production of 4′-OHDF and 5-OHDF in zebrafish.

In contrast to the above results, 6β-hydroxytestosterone, which is a metabolite of testosterone via human CYP3A4/5, was not detected at 50 or 120 hpf in zebrafish embryos or larvae during exposure to testosterone. This absence of detection is consistent with previous data on zebrafish embryos and larvae at up to 120 hpf and in zebrafish adults [22]. Conversely, other studies have detected 6β-hydroxytestosterone in zebrafish larvae at 7 dpf or in zebrafish adults [2,46]. Chng et al. detected seven mono-hydroxylated metabolites of testosterone, including 6β-hydroxytestosterone, in zebrafish [2]. We surmise that the differences in detection of 6β-hydroxytestosterone were caused by differences in experimental methods or conditions, including the duration of microsome incubation with the substrate, the stage of use of embryos or larvae, and the zebrafish strain. More detailed studies are required to clarify the metabolism of testosterone in zebrafish embryos, larvae, and adults.

The activity of the Cyp3A subfamily in zebrafish remains controversial. Here, in zebrafish embryos and larvae, we found differences in the generation of 5-OHDF and 6β-hydroxytestosterone, which are metabolites via the same CYP3A4/5 in humans. 3-Methoxymorphinan, which is a metabolite of dextromethorphan via CYP3A4/5 in humans, has been detected in adult zebrafish liver microsomes, although no metabolites of this compound have been detected in the microsomes from whole embryos or larvae at 5, 24, 48, 72, 96, or 120 hpf [22]. Neither 1-hydroxymidazolam nor 4-hydroxymidazolam, which are metabolites of midazolam via CYP3A4/5 in humans, has been detected in adult zebrafish liver microsomes or embryonic or larval microsomes or homogenates at up to 7 dpf [22,46]. These data suggest that there are differences in substrate specificity between humans and zebrafish. Moreover, even if human CYP3A4/5-mediated metabolites are detectable in zebrafish embryos or larvae or adults, among developmental stages there may be differences in the intensity of the metabolic activity of the homologous zebrafish isoform.

In summary, we profiled the mRNA expression levels, organ distribution, and metabolic activity of various drug-metabolizing CYPs in developing zebrafish embryos and larvae. We suggest that (1) many metabolic CYP subtypes are expressed before full liver development, and the transcription levels of some of these subtypes in whole zebrafish embryos or larvae are comparable to, or even higher than, those in the adult liver; (2) early zebrafish embryos and larvae may have metabolic functions similar to those of human CYP isoforms (CYP1A2, 2C9, and 3A4/5) without depending on development of the liver; and (3) there may be differences between humans and zebrafish in the substrate specificity of some CYP isoforms, as was the case here for human CYP3A4/5-mediated metabolites. In conclusion, from our results, we consider that the zebrafish is a suitable model for use in the drug discovery process, especially for drugs with toxic metabolites, and that the metabolic activity in developing zebrafish need to be clarified in detail for more precise assessment of effects of drugs.

## 4. Materials and Methods

### 4.1. Test Organisms and Collection of Fertilized Eggs

Wild-type zebrafish (*Danio rerio*, NIES-R strain, National Institute for Environmental Studies, Tsukuba, Japan) were used to study metabolic activity and for qPCR assay for CYP subtypes. The long-fin strain (maintained in our lab) was used for in situ hybridization. Maintenance of test organisms and collection of fertilized eggs were performed as described previously [10].

### 4.2. Exposure of Zebrafish Embryos and Larvae to Test Compounds

For the in situ hybridization studies and qPCR, 10 zebrafish embryos were exposed individually to 100 µM β-naphthoflavone (Sigma-Aldrich, St. Louis, MO, USA), 1.0 ppb TCDD (Cambridge Isotope Laboratories, Andover, MA, USA), 100 µM rifampicin (FUJIFILM Wako Pure Chemical, Osaka, Japan), or 100 µM phenobarbital sodium (Tokyo Kasei, Tokyo, Japan) in 3 mL fish water (38.7 mM NaCl, 1.0 mM KCl, 1.7 mM HEPES-NaOH pH 7.2, 2.4 mM CaCl_2_) in a plastic Petri dish during 30 hpf to 55 hpf. Embryos were dechorionized, except in the case of those exposed to TCDD due to high transparency. To study the metabolic activity of embryonic and larval zebrafish, caffeine, diclofenac, testosterone, acetone, acetonitrile, and methanol were purchased from FUJIFILM Wako Pure Chemical. 1,7-Dimethylxanthine, 4′-OHDF, 5-OHDF, and 6β-hydroxytestosterone were obtained from Sigma-Aldrich. DMSO was purchased from Nacalai Tesque (Kyoto, Japan).

To calculate the content of each compound per one whole zebrafish embryo or larva (i.e., the C_e_), we first determined the maximum test solution concentrations that caused neither morphological nor behavioral abnormalities during exposure. Those test solution concentrations were 10 mg/L for caffeine, 3.2 mg/L for diclofenac, and 3.0 mg/L for testosterone. Each test solution was prepared with dilution water only (reconstituted water: ISO 6341-1982) [51] in the case of caffeine and diclofenac and with dilution water containing 0.01% DMSO in the case of testosterone. Beginning at 5 hpf, normal embryos were exposed to the test solution (>1 mL/embryo) in a glass container. The plastic lid of the container was kept closed, and the embryos and larvae were maintained for up to 120 hpf at 28 ± 1 °C on a 14-h light–10-h dark cycle. The test solution of caffeine and that of diclofenac were not renewed. The testosterone solution was renewed at 48 hpf to maintain the concentration of testosterone in the water.

### 4.3. qPCR

qPCR analysis was performed to examine the expression levels of various CYP subtypes [26]. Total RNA was extracted from embryos, larvae and livers of adult fishes of both sexes with TRI-Reagent (Sigma-Aldrich), and cDNA was prepared with a ReverTra Ace qPCR kit (Toyobo, Osaka, Japan) in accordance with the manufacturer’s instructions. We used between 15 and 20 embryos or larvae for each cDNA sample, and we prepared six cDNA samples for each developmental stage (*n* = 6). We prepared 12 cDNA samples from 12 livers of adult fish (six from males and six from females). qPCR analysis was performed with a LightCycler480 System II (Nippon Genetics, Tokyo, Japan) by using Thunderbird qPCR Mix (Toyobo). The primer sets used for qPCR analysis are listed in Appendix A.

All primer sets were confirmed to produce a single peak in the melting curve, as well as a single band by agarose gel electrophoresis in a preliminary study.

### 4.4. Whole-Mount In Situ Hybridization

Whole-mount in situ hybridization was performed as described previously [52]. Paraformaldehyde-fixed embryos were hybridized with digoxigenin-incorporated antisense RNA probes of CYP subtypes at 65 °C overnight (DIG (digoxigenin) RNA Labeling Kit, Sigma-Aldrich). To obtain the CYP probes, CYP DNAs were amplified by PCR reaction with KOD FX Neo enzyme solution (Toyobo) with the primer sets listed in Appendix A. This was followed by subcloning to pTAC2 TA vector (BioDynamics, Tokyo, Japan) and in vitro transcription with SP6 RNA polymerase (New England Biolabs, Ipswich, MA, USA). Following hybridization and washing, the embryos were incubated with anti-DIG antibody conjugated with alkaline phosphatase (Sigma-Aldrich) at 4 °C overnight. The color reaction was performed by incubation in BM Purple substrate or Fast Red (Sigma-Aldrich).

### 4.5. Measurement of Test Compound Content per One Whole Zebrafish Embryo or Larva Every 24 hpf

Exposed embryos (three to 20 embryos or larvae per replication, three replicates per level per time point) were analyzed every 24 hpf up to a maximum of 120 hpf. The chorions of embryos were removed with forceps under a microscope before sampling for the C_e_ measurements. The dechorionated embryos (24 hpf) or the larvae (from 48 hpf onwards) were individually transferred through four beakers (for about 10 s per beaker) containing 500 mL of fresh dechlorinated tap water to remove chemical residues on the body surface. After that step, the embryos or larvae were homogenized with a silicone pestle in a 1.5 mL sampling tube containing a mixed solvent of the same composition as each liquid chromatography eluent to extract the test compound. The dechorionated embryos or the larvae were homogenized to extract the test compound. The mixture was centrifuged at 20,000× *g* for 10 min at 10 °C with a refrigerated centrifuge (CR21N, Hitachi Koki, Tokyo, Japan). The supernatant was collected in a volumetric flask. The extraction procedure was performed twice, and the supernatants from the two batches were mixed and brought up to a volume of 1 mL with the mixed solvent of the same composition as each liquid chromatography eluent to extract the test compound. They were then filtered through a Millex-LG membrane filter with a 0.2 μm pore size (Merck KGaA, Darmstadt, Germany). Standard solutions of each test compound without adding mixture of embryos or larvae in the control were used to make each calibration curve, because no matrix effect was recognized.

For caffeine and testosterone, the treated samples were analyzed by high-performance liquid chromatography by using an LC-20AD solvent delivery system (Shimadzu, Kyoto, Japan) equipped with an L-column2 ODS (octadecyl–silica) column (length, 150 mm; inner diameter, 4.6 mm for caffeine or 2.1 mm for testosterone; particle size, 5 µm; Chemicals Evaluation and Research Institute, Tokyo, Japan). Caffeine and testosterone were detected by an SPD-20AV UV-VIS (ultraviolet–visible light) detector (Shimadzu). Samples (50 μL for caffeine; 20 μL for testosterone) were eluted in each mobile phase of methanol: water = 2:8 *v*/*v* for caffeine and acetonitrile: water = 1:1 *v*/*v* for testosterone. Flow rates were 1.00 mL/min for caffeine and 0.20 mL/min for testosterone. These compounds were detected by UV light of different wavelengths (270 nm for caffeine and 241 nm for testosterone).

For diclofenac, treated samples were measured by liquid chromatography–tandem mass spectrometry (LC-MS/MS) analysis with an LCMS-8060 triple quadrupole mass spectrometer (Shimadzu) and a Nexera X2 ultra high-performance liquid chromatograph (Shimadzu) equipped with an ACQUITY UPLC BEH C18 column (length, 50 mm; inner diameter, 2.1 mm; particle size, 1.7 µm; Nihon Waters, Tokyo, Japan). Each 10 μL sample was eluted at a flow rate of 0.30 mL/min and in the following gradient mobile phase of water with 0.1% formic acid (A) and acetonitrile with 0.1% formic acid: 0 to 1 min 80% A; 1 to 35 min 80% to 65% A; and 35 to 45 min 65% A. Diclofenac was monitored by using an electrospray ionization (ESI) probe. Data were acquired in positive ion mode by using multiple reaction monitoring (MRM). The temperatures applied were, for the auto sampler 5 °C, column 40 °C, interface 300 °C, desolvation 526 °C, desolvation line 240 °C, and heat block 400 °C. The flow rates were, for the nebulizer gas 1.50 L/min, heating gas 10.00 L/min, and drying gas 10.00 L/min. Three transitions were monitored for diclofenac: the precursor ion *m*/*z*, product ion *m*/*z*, Q1 pre-bias, collision energy, and Q3 pre-bias were, respectively, (1) 296.10, 215.10, −11.0 V, −19.0 V, and −22.0 V; (2) 296.10, 250.00, −11.0 V, −13.0 V, and −26.0 V; and (3) 296.10, 278.00, −14.0 V, −10.0 V, and −30.0 V.

In these experiments, the concentrations of each test compound in the water (C_w_s) were measured. For caffeine and diclofenac, C_w_s were measured at the start and end of the exposure period. For testosterone, the test solutions were analyzed in two sets: (1) solutions freshly prepared at the start of exposure and at 48 hpf and (2) solutions before renewal at 48 hpf and at the end of the exposure period. Sampled solutions were diluted with a solution of the same composition as the eluent for each compound. The equipment and conditions for measuring the C_w_ values of caffeine, diclofenac, and testosterone were the same as those used to measure their C_e_ values.

### 4.6. Measurement of Test Compound Metabolite Contents in Zebrafish Embryos and Larvae

To measure the C_e_ values of the target metabolites of test compounds during exposure, larvae were analyzed at 50 hpf and 120 hpf in the case of 1,7-dimethylxanthine and 6β-hydroxytestosterone, and embryos or larvae were analyzed every 24 hpf until 120 hpf in the case of 4′-OHDF and 5-OHDF.

For 1,7-dimethylxanthine and 6β-hydroxytestosterone, exposed larvae (200 to 1000 larvae per replication, three replicates per level per time point for 1,7-dimethylxanthine, one replicate per level per time point for 6β-hydroxytestosterone) were collected into a centrifuge tube containing acetone and homogenized by using a Polytron homogenizer (PT3100, Kinematica AG, Lucerne, Switzerland) to extract the test compound. Then, the mixture was centrifuged at 20,000× *g* for 15 min at 10 °C with a refrigerated centrifuge (CR21N, Hitachi Koki). The supernatant was collected in a sampling tube. The extraction procedure was performed twice. The supernatants of two batches were mixed and the mixture was evaporated completely by nitrogen gas spraying. A solution (500 μL) of the same composition as the eluent for each compound was added to the tube, and then the solution was filtered through a Millex-LG membrane filter with a 0.2-μm pore size (Merck). The filtrate was double diluted with a solution of the same composition as the eluent for each compound, and the resulting solution was mixed to prepare the sample for measurement. For 1,7-dimethylxanthine, to standardize the content of the larval matrix, a mixture of larvae in the control or the vehicle control was added to each standard solution in a stepwise concentration to create each calibration curve.

For 1,7-dimethylxanthine, the treated samples were analyzed by LC-MS with an LCMS-8060 (Shimadzu) and a Nexera X2 (Shimadzu) equipped with an ACQUITY UPLC BEH C18 column (length, 50 mm; inner diameter, 2.1 mm; particle size, 1.7 µm; Nihon Waters). Each 5 μL sample was eluted in a mobile phase of methanol: water = 2:8 at a flow rate of 0.20 mL/min. 1,7-dimethylxanthine was monitored by using an ESI probe. Data were acquired in positive ion mode by using MRM. The temperatures applied were, for the auto sampler 5 °C, column 40 °C, interface 300 °C, desolvation line 240 °C, and heat block 400 °C. The flow rates were, for the nebulizer gas 1.50 L/min, heating gas 10.00 L/min, and drying gas 10.00 L/min. One transition was monitored for 1,7-dimethylxanthine: the precursor ion *m*/*z*, product ion *m*/*z*, Q1 pre-bias, collision energy, and Q3 pre-bias were respectively 181.20, 124.20, −11.0 V, −20.0 V, and −12.0 V.

For 6β-hydroxytestosterone, the treated samples were analyzed by LC-MS with an LCMS-8060 (Shimadzu) and a Nexera X2 (Shimadzu) equipped with an ACQUITY UPLC BEH C18 column (length, 50 mm; inner diameter, 2.1 mm; particle size, 1.7 µm; Nihon Waters). Each 10 μL sample was eluted in a mobile phase of acetonitrile (with 0.1% formic acid): water (with 0.1% formic acid) = 3:7 at a flow rate of 0.20 mL/min. 6β-hydroxytestosterone was monitored by using an ESI probe. Data were acquired in positive ion mode by using MRM. The temperatures applied were, for the auto sampler 5 °C, column 40 °C, interface 300 °C, desolvation line 240 °C, and heat block 400 °C. The flow rates were, for the nebulizer gas 1.50 L/min, heating gas 10.00 L/min, and drying gas 10.00 L/min. Two transitions were monitored in the case of 6β-hydroxytestosterone: the precursor ion *m*/*z*, product ion *m*/*z*, Q1 pre-bias, collision energy, and Q3 pre-bias were respectively (1) 305.00, 269.40, −21.0 V, −16.0 V, and −18.0V; and (2) 305.00, 287.30, −22.0 V, −15.0 V, and −29.0 V.

For 4′-OHDF and 5-OHDF, LC-MS/MS was used to analyze the same samples as treated in the experiment for measuring the C_e_ values of diclofenac, without dilution. Standard control solutions without addition of the mixture were used to make each calibration curve, because no matrix effect was recognized. The equipment and conditions, with the exception of the target ion for monitoring, were the same as used to measure the C_e_ values of diclofenac. The same three transitions were monitored for 4′-OHDF and 5-OHDF: the precursor ion *m*/*z*, product ion *m*/*z*, Q1 pre-bias, collision energy, and Q3 pre-bias were respectively (1) 312.10, 231.10, −22.0 V, −21.0 V, and −24.0V; (2) 312.10, 266.00, −16.0 V, −13.0 V, and −17.0 V; and (3) 312.10, 294.00, −16.0 V, −11.0 V, and −13.0 V.

### 4.7. Quantification of Analytical Samples

A calibration curve for each target substance (regression equation by using the least-squares method: *Y* = *aX* + *b*, where *Y* is the analytical response and *X* is the concentration of the target substance) was made by using four or five concentrations of standard solution. We confirmed that each calibration curve met the following criteria: (i) The correlation coefficient (*r*) was >0.995; and (ii) the absolute value of the intercept (*b*) was within 5% of the maximum analytical response and the linear regression line was treated as a straight line from the origin. Then, all target substances were quantified by using the absolute calibration curve method and one concentration of standard solution. For all substances except 4′-OHDF and 5-OHDF, the peak area obtained from each chromatogram was treated as the analytical response. Because the abovementioned conditions did not allow us to separate 4′-OHDF and 5-OHDF chromatographically, each peak height obtained from the chromatogram was treated as the analytical response for both 4′-OHDF and 5-OHDF.

### 4.8. Statistics

In the qPCR experiment, the results are presented as means ± SEM (*n* = 6). Expression levels were compared between the respective control and the comparison groups by using one-way ANOVA followed by Dunnett’s test by means of Statcel 4 (OMS Publishing, Higashikurume, Japan) (*p* < 0.05). Results from the measurement of C_e_ values are presented as means ± SD (*n* = 3), and significant differences between two specific C_e_ values were determined by two-tailed *t*-test (*p* < 0.05) by means of SPSS statistics version 22 (IBM, Armonk, NY, USA).

## Figures and Tables

**Figure 1 pharmaceuticals-13-00456-f001:**
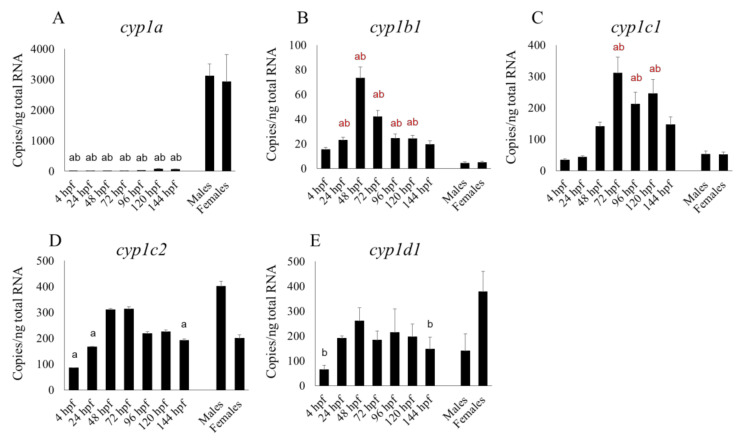
Developmental expression of five mRNAs of *cyp1* isoforms in zebrafish whole embryos and larvae from 4 to 144 h post-fertilization (hpf) in comparison with their expression in the livers of adults of both sexes. Results of *cyp1a* (**A**), *cyp1b1* (**B**), *cyp1c1* (**C**), *cyp1c2* (**D**) and *cyp1d1* (**E**) are shown. “a” indicates a significant (*p* < 0.05) difference compared with male livers and “b” is a significant (*p* < 0.05) difference compared with female livers. Black letters indicate significantly lower expression in embryos or larvae compared with adult livers, and red letters indicate significantly higher expression in embryos or larvae compared with adult livers. *n* = 6.

**Figure 2 pharmaceuticals-13-00456-f002:**
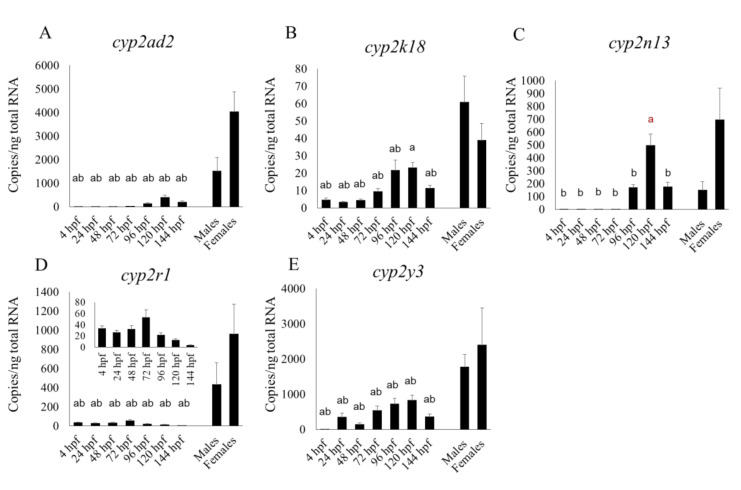
Developmental expression of five mRNA of *cyp2* isoforms in zebrafish whole embryos and larvae from 4 to 144 h post-fertilization (hpf) in comparison with their expression in the livers of adults of both sexes. Results of *cyp2ad2* (**A**), *cyp2k18* (**B**), *cyp2n13* (**C**), *cyp2r1* (**D**) and *cyp2y3* (**E**) are shown. An inset graph was added in embryonic and larval *cyp2r1* expressions (**D**). “a” indicates a significant (*p* < 0.05) difference compared with male livers and “b” is a significant (*p* < 0.05) difference compared with female livers. Black letters indicate significantly lower expression in embryos or larvae compared with adult livers, and red letters indicate significantly higher expression in embryos or larvae compared with adult livers. *n* = 6.

**Figure 3 pharmaceuticals-13-00456-f003:**
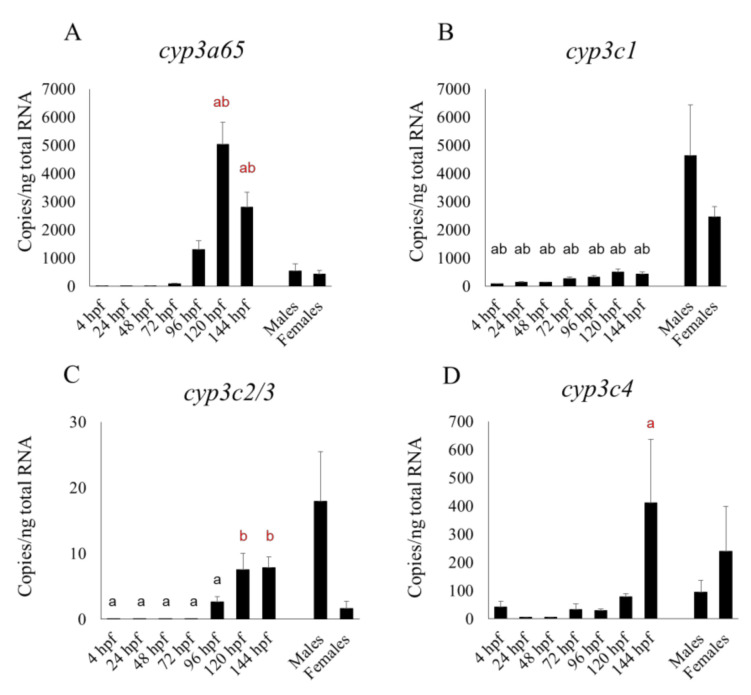
Developmental expression of four mRNA of *cyp3* isoforms　in zebrafish whole embryos and larvae from 4 to 144 h post-fertilization (hpf) in comparison with their expression in the livers of adults of both sexes. Results of *cyp3a65* (**A**), *cyp3c1* (**B**), *cyp3c2/3* (**C**) and *cyp3c4* (**D**) are shown. “a” indicates a significant (*p* < 0.05) difference compared with male livers and “b” is a significant (*p* < 0.05) difference compared with female livers. Black letters indicate significantly lower expression in embryos or larvae compared with adult livers, and red letters indicate significantly higher expression in embryos or larvae compared with adult livers. *n* = 6.

**Figure 4 pharmaceuticals-13-00456-f004:**
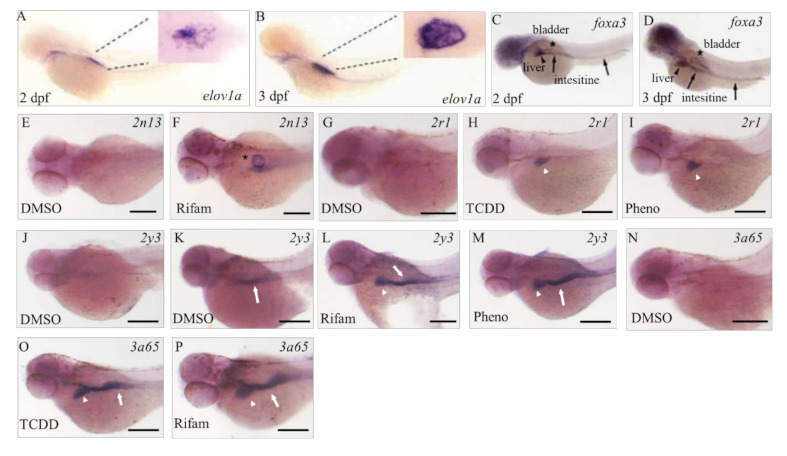
Spatial expression of mRNAs of CYP subtypes. Zebrafish embryos were exposed to 100 µM rifampicin (Rifam; (**F**,**L**,**P**)), 100 µM phenobarbital (Pheno; (**I**,**M**)), or 1.0 ppb TCDD (**H**,**O**) and 0.1% dimethyl sulfoxide (DMSO) as vehicle control (DMSO; (**E**,**G**,**J**,**K**,**N**)). (**A**,**B**) are published images of *elov1a*, a marker of the primordial bladder ((**A**) at 2 days post-fertilization [dpf], (**B**) at 3 dpf) [37]. (**C**,**D**) are published images of *foxa3*, a marker of the primordial liver (black arrowheads) and intestine (black arrows) and bladder (asterisks) ((**C**) at 2 dpf, (**D**) at 3 dpf) [38]. Larval zebrafish (55 hpf) were used for whole-mount in situ hybridization with RNA probes for *cyp2n13* (**E**,**F**), *cyp2r1* (**G**–**I**), *cyp2y3* (**J**–**M**), and *cyp3a65* (**N**–**P**). Asterisks indicate possible bladder primordium (**F**) and white arrowheads and arrows indicate possible primordia of the liver and intestine, respectively (**H**–**P**). Scale bars in (**E**–**P**) indicate 200 µm. (**A**–**D**) are reproduced from previous reports ([37] for (**A**,**B**), and [38] for (**C**,**D**)).

**Figure 5 pharmaceuticals-13-00456-f005:**
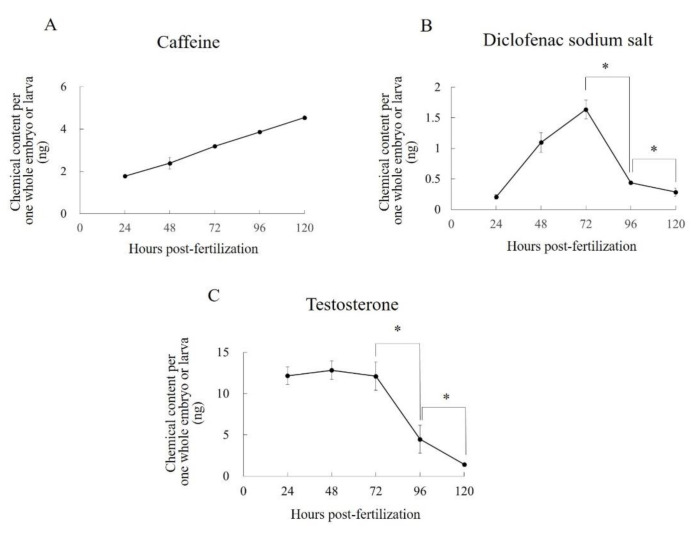
Contents of (**A**) caffeine, (**B**) diclofenac sodium salt, and (**C**) testosterone per one whole embryo/larva during exposure to each test compound. Each plot shows the mean values of three replicates. Each error bar shows the standard deviation of the replicates. Significant differences between two specific data points, as determined by a two-tailed *t*-test, are shown by an asterisk (*p* < 0.05). Statistical analyses were performed only for the specific sets of data points shown by the asterisks.

**Figure 6 pharmaceuticals-13-00456-f006:**
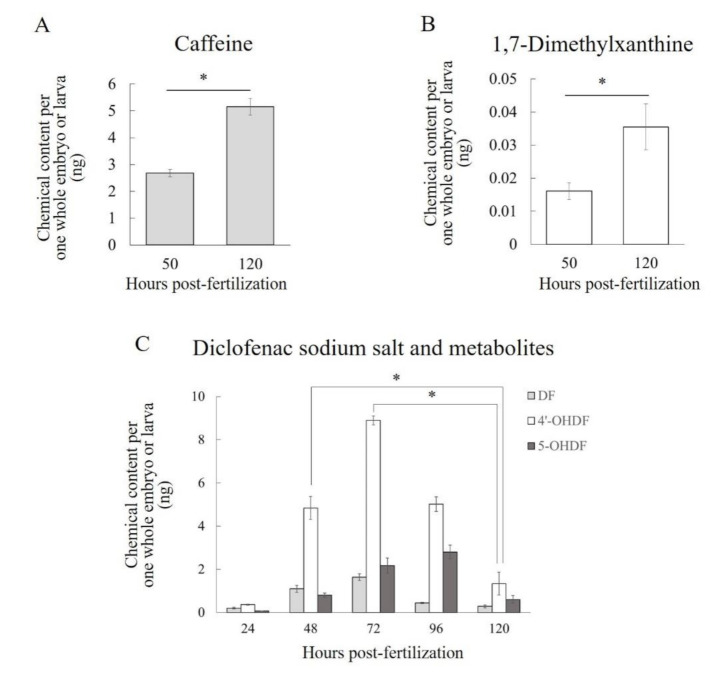
Contents of (**A**) caffeine, (**B**) 1,7-dimethylxanthine, and (**C**) diclofenac sodium salt (DF), 4′-hydroxydiclofenac (4′-OHDF), and 5-hydroxydiclofenac (5-OHDF) per one whole embryo or larva during exposure to each test compound. Each bar shows the mean value of three replicates. Each error bar shows the standard deviation of the replicates. Significant differences between two specific data points, as determined by a two-tailed *t*-test, are shown by asterisks (*p* < 0.05). Statistical analyses were performed only for the specific sets of data points shown by the asterisks.

**Table 1 pharmaceuticals-13-00456-t001:** Measured concentrations of test compounds in test solutions (C_w_s) used to determine concentrations in zebrafish embryos or larvae (C_e_s).

Caffeine		Diclofenac Sodium Salt		Testosterone	
Test Level	Time-Weighted Mean of Measured Concentration	Test Level	Time-Weighted Mean of Measured Concentration	Test Level	Time-Weighted Mean of Measured Concentration
10	10.2 (102%)	3.2	3.41 (107%)	3.0	2.80 (93.3%)

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
