# Peer review of "Cytochrome P450 Expression and Chemical Metabolic Activity before Full Liver Development in Zebrafish"

_pharmaceuticals, 2020, doi:10.3390/ph13120456_

Round 1

Reviewer 1 Report

In the manuscript “Cytocrome P450 expression and chemical metabolic activity before full liver development in zebrafish”, Nawaji T. ad colleagues characterize the expression of 14 CYP subtypes, and the metabolic activity of a set of human CYP substrates in zebrafish. The study shows that many CYP subtypes are expressed and may have metabolic activity in zebrafish embryos before full liver development.

Overall, the study is well described and technically sounds.

Major points:

Early expression (4 hpf – 24 hpf) of considered CYP types is analyzed exclusively via quantitative PCR. The authors should explain why a parallel analysis via whole mount in situ hybridization (WISH) has not been performed. By WISH, it is possible not only to have a confirmation of the qPCR results through an independent technique, but also to appreciate which parts (endoderm, ectoderm, mesoderm) of the zebrafish embryos at blastula, gastrula or somitogenesis stage are expressing the different CYP types.

WISH experiments lack appropriate negative controls. As negative controls, the use of sense probes is usually recommended, to verify that sense probes do not recognize any target, thus corroborating antisense probes specificity. These experiments should be performed and, in case, displayed as supplementary materials.

Minor points:

Throughout the whole manuscript, including figures, I would recommend to check the gene and protein names, to verify that they follow the international conventions, as mentioned in the ZFIN database:

https://wiki.zfin.org/display/general/ZFIN+Zebrafish+Nomenclature+Conventions

Introduction, line 35: it is not correct to state that until 120 hpf zebrafish embryos are “non-animals”.

Zebrafish embryos are always animals, under a scientific and biological point of view. Before 120 hpf, zebrafish embryos/larvae are “at stages not protected by animal testing laws”, or, alternatively, “non-protected animals”, based on current animal testing laws.

Introduction, line 66: just as a matter of taste, I would suggest substituting “insufficient” with “limited”.

Figure 1, Figure 2, Figure 3: lettering of the different panels seems a bit useless, if it is not mentioned neither in the figure legends, nor in the main text.

Figure 4: quality of the imaging in panels E-N is a bit low, and background colors rather heterogeneous among each other. Scale bars are missing.

Materials and methods, lines 353-354:  repetition of “in a plastic Petri dish”.

Note: Petri should have the P in uppercase.

Materials and methods, line 373: how many embryos/larvae or adult organs were on average used for each qPCR sample?

Material and methods, line 380 and following ones on Whole mount in situ hybridization:

Preparation and use of appropriate negative controls (sense probes) is missing.

Materials and methods, line 501 and following ones on Statistics:

A statement on sample size, blinded experiments, in duplicate or in triplicate, and so on, is missing.

Author Response

Major points:

Point 1: Early expression (4 hpf – 24 hpf) of considered CYP types is analyzed exclusively via quantitative PCR. The authors should explain why a parallel analysis via whole mount in situ hybridization (WISH) has not been performed. By WISH, it is possible not only to have a confirmation of the qPCR results through an independent technique, but also to appreciate which parts (endoderm, ectoderm, mesoderm) of the zebrafish embryos at blastula, gastrula or somitogenesis stage are expressing the different CYP types.

Response 1: We agree with the Reviewer’s suggestion is very important and attractive. However, we confirmed metabolic activity of diclofenac after 48 hpf, but not in 24 hpf. As our major interest in this study is to confirm functional metabolic CYP expressions in developing zebrafish, we focused expressions of some CYP subtypes with 55 hpf embryos for WISH in this study. We would like to try it in the following studies.

Point 2: WISH experiments lack appropriate negative controls. As negative controls, the use of sense probes is usually recommended, to verify that sense probes do not recognize any target, thus corroborating antisense probes specificity. These experiments should be performed and, in case, displayed as supplementary materials.

Response 2: As suggested by the Reviewer, use of sense probe is one of the representative methods as a negative control. We believe that ideal negative control for WISH with developing zebrafish is difficult to obtain. Although sense probe is equal to antisense probe in the light of Tm values, these two probes showed different selectivity to endogenous mRNA in tissues in some cases. In the present experiments, however, we did not find positive signals in vehicle-control embryos/larvae. We described that we hardly (only cyp2y3) or never (the others) observe positive signal with four CYP antisense probes studied with vehicle-treated control larvae. Positive signals of cyp2y3 were very rarely observed in control larvae. On the contrary, we were able to observe positive signals with those antisense probes with many larvae exposed by rifampicin, phenobarbital and TCDD. We believe that this fact tells that our WISH study is specific and reliable. We added typical images of vehicle-treated control larvae in Figure 4 in the revised manuscript. Also, we added a short explanation in the revised manuscript.

Minor points:

Point 1: Throughout the whole manuscript, including figures, I would recommend to check the gene and protein names, to verify that they follow the international conventions, as mentioned in the ZFIN database:

https://wiki.zfin.org/display/general/ZFIN+Zebrafish+Nomenclature+Conventions

Response 1: We checked nomenclature and corrected the respective names as suggested by the Reviewer.

Point 2: Introduction, line 35: it is not correct to state that until 120 hpf zebrafish embryos are “non-animals”. Zebrafish embryos are always animals, under a scientific and biological point of view. Before 120 hpf, zebrafish embryos/larvae are “at stages not protected by animal testing laws”, or, alternatively, “non-protected animals”, based on current animal testing laws.

Response 2: We agree with the Reviewer’s suggestion. We revised the sentence as follows: “because in Europe zebrafish embryos are considered non-protected animals until the stage of independent feeding at 120 h post-fertilization (hpf) based on directive on the protection of animals used for scientific purposes, ”

Point 3: Introduction, line 66: just as a matter of taste, I would suggest substituting “insufficient” with “limited”.

Response 3: We corrected it as suggested.

Point 4: Figure 1, Figure 2, Figure 3: lettering of the different panels seems a bit useless, if it is not mentioned neither in the figure legends, nor in the main text.

Response 4: We added lettering of Figure panels in the revised manuscript.

Point 5: Figure 4: quality of the imaging in panels E-N is a bit low, and background colors rather heterogeneous among each other. Scale bars are missing.

Response 5: Although signals were relatively week in these early larvae even with induction, we tried it again and improved images as we could, as well as scale bars.

Point 6: Materials and methods, lines 353-354: repetition of “in a plastic Petri dish”.

Note: Petri should have the P in uppercase.

Response 6: We corrected it as suggested.

Point 7: Materials and methods, line 373: how many embryos/larvae or adult organs were on average used for each qPCR sample?

Response 7: We are very sorry. We used between 15 and 20 embryos or larvae for each cDNA sample, and we prepared six cDNA samples for each developmental stage (n = 6). We prepared 12 cDNA samples from 12 livers of adult fish (six from males and six from females). We added these description in Materials and methods and Figure legends.

Point 8: Material and methods, line 380 and following ones on Whole mount in situ hybridization: Preparation and use of appropriate negative controls (sense probes) is missing.

Response 8: This is the same comments as Major point No. 2. We added images of negative controls in Figure 4 and short explanation in the revised manuscript to explain that positive WISH signals were recognized in larvae that were treated with inducers almost exclusively.

Point 9: Materials and methods, line 501 and following ones on Statistics:

A statement on sample size, blinded experiments, in duplicate or in triplicate, and so on, is missing.

Response 9: We added required information in the revised manuscript.

Reviewer 2 Report

The manuscript discussed the cytochrome P450 expression and activity in Zebrafish before full liver development.  The paper is clearly written and easy to follow.  The approaches are pretty standard.  Here are the comments for the authors to consider:

  1. Abbreviations are explained in text when mentioned for the first time, but I lack the abbreviations list. Abbreviations list should be added because abbreviations (ce, 5-OHDF, ...) may be unfammiliar to reader.  
  2. mRNA expression was studied. Do you have any idea about protein expression? Correspond mRNA level to protein level?
  3. Is it known which cytochrome P450 are responsible for production of studied metabolites in Zebrafish?
  4. Is Zebrafish a suitable model in drug discovery process pursuant to your results?

Author Response

Point 1: Abbreviations are explained in text when mentioned for the first time, but I lack the abbreviations list. Abbreviations list should be added because abbreviations (ce, 5-OHDF, ...) may be unfammiliar to reader.

Response 1: We added list of abbreviations in the revised manuscript.

Point 2: mRNA expression was studied. Do you have any idea about protein expression? Correspond mRNA level to protein level?

Response 2: Reviewer’s suggestion is very important. Expressions of mRNA of most major metabolic CYP isoforms almost reflect amounts of proteins except CYP2Es in the human liver (Drozdzik et al., 2018). We think that mRNA levels reflect protein levels at least to some extent. However, some CYP isoforms including CYP2Es, showing higher amount of mRNAs but minor amount of protein in mammals (Mohri et al., 2010). As antibodies specific to zebrafish CYP metabolic CYP isoforms are not available except CYP1A and CYP3C1 at the moment, we were not able to compare amount of endogenous proteins of most CYP isoforms. We added a paragraph on this discussion in the revised manuscript.

Drozdzik, M.; Busch, D.; Lapczuk, J.; Müller, J.; Ostrowski, M.; Kurzawski, M.; Oswald, S. Protein abundance of clinically relevant drug transporters in the human liver and intestine: a comparative analysis in paired tissue specimens. Clin. Pharmacol. Ther. 2018, 104, 515–524.

Mohri, T.; Nakajima, M.; Fukami, T.; Takamiya, M.; Aoki, Y.; Yokoi, T. Human CYP2E1 is regulated by miR-378. Biochem. Pharmacol. 2010, 79, 1045–1052.

Point 3: Is it known which cytochrome P450 are responsible for production of studied metabolites in Zebrafish?

Response 3: We exactly anxious to know that point. However, metabolic properties of most CYP isoforms in zebrafish are unknown, especially for CYP2 isoforms. Although some CYP1 isoforms, CYP3A65 and CYP3C1 were relatively studied with yeast expression system, only limited information is available on metabolic with some substrates (Corley-Smith et al., 2006; Scornaienchi et al., 2010; Stegeman et al. 2015). As mentioned in L306-309 in the original manuscript, we would like to determine the CYP isoform that is responsible for metabolism of diclofenac and caffeine using heterologous zebrafish CYP expression system in E. coli or yeast in the future experiments. We added a sentence that metabolic properties of most CYP isoforms in zebrafish are unknown, especially for metabolism of caffeine and diclofenac.

Corley-Smith, G.E.; Su, H.T; Wang-Buhler, J.L.; Tseng, H.P.; Hu, C.H.; Hoang, T.; Chung, W.G.; Buhler, D.R. CYP3C1, the first member of a new cytochrome P450 subfamily found in zebrafish (Danio rerio). Biochem. Biophys. Res. Commun. 2006, 340, 1039–1046.

Scornaienchi, M.L.; Thornton, C.; Willett, K.L.; Wilson, J.Y. Cytochrome P450-mediated 17beta-estradiol metabolism in zebrafish (Danio rerio). J. Endocrinol. 2010, 206, 317–25.

Stegeman, J.; Behrendy, L.; Woodin, B.R.; Kubota, A.; Lemaire, B.; Pompon, D.; Goldstone, J.V.; Urban, P. Functional characterization of zebrafish cytochrome P450 1 family proteins expressed in yeast. Biochim. Biophys. Acta, Gen. Subj. 2015, 1850, 2340–2352.

Point 4: Is Zebrafish a suitable model in drug discovery process pursuant to your results?

Response 4: We believe so. Our results show that larval zebrafish possess metabolic CYP isoforms that show similar metabolic properties with human, although we studied only two substrates. Additionally, we provide transcriptional profiles of 14 metabolic CYP isoforms at early stages, in which significant metabolism of these two substrates were confirmed. We added a conclusion sentence in the end of Discussion of the revised manuscript. Thank you so much.

Others: As we quoted Hakkola et al. 1998. Crit. Rev. Toxicol. [40] by mistake, we replaced it with Hakkola et al. 1998. Pharmacol. Toxicol.

Round 2

Reviewer 1 Report

No more comments.